# COVID-19’s Impact on the Pan African Sanctuary Alliance: Challenging Times and Resilience from Its Members

**DOI:** 10.3390/ani13091486

**Published:** 2023-04-27

**Authors:** Nora Bennamoun, Marco Campera, Gregg Tully, K.A.I. Nekaris

**Affiliations:** 1School of Social Sciences, Oxford Brookes University, Oxford OX3 0BP, UK; norabennamoun2060@gmail.com (N.B.);; 2Department of Biological and Medical Sciences, Oxford Brookes University, Oxford OX3 0BP, UK; 3Pan African Sanctuary Alliance (PASA), Portland, OR 97219, USA; gregg@savethedogs.eu

**Keywords:** SARS-CoV-2, conservation, sanctuary, primates, management, ecotourism, best practices, pandemic

## Abstract

**Simple Summary:**

Sanctuaries have important roles in the in situ conservation of wildlife, including research (e.g., the monitoring of populations) and applied conservation (e.g., translocations). As a consequence of the COVID-19 pandemic, many sanctuaries have been impacted, especially due to the reduction in income from ecotourism. We analyzed the responses of the 23 sanctuary members of the Pan African Sanctuary Alliance (PASA) to the COVID-19 pandemic, analyzing the periods before, at the start of, and during the pandemic in terms of primates rescued, employees, and expenses. Overall, sanctuaries managed to continue their activities despite the significant limitation to their incomes. We suggest the best measures to be taken to mitigate the post-pandemic effects and to prevent future outbreaks.

**Abstract:**

The worldwide pandemic caused by SARS-CoV-2 challenged conservation organizations. The lack of tourism has benefited or negatively affected wildlife organizations in various ways, with several primate sanctuaries struggling to cope with the COVID-19 crisis and to keep providing for their inhabitants. In addition, the genetic similarity between great apes and humans puts them at higher risk than any other species for the transmission of COVID-19. PASA is a non-profit organization comprising 23 sanctuaries, and cares for many species of primate, including African great apes. In light of the pandemic, we aimed to understand the direct effects of COVID-19 on PASA management throughout three time periods: before (2018–2019), at the start of (2019–2020), and during (2020–2021) the pandemic. We collected data via annual surveys for PASA members and ran Generalized Linear Mixed Models to highlight any significant differences in their management that could be linked to COVID-19. Our findings demonstrated no particular impact on the number of primates rescued, employees, or expenses. However, revenues have been decreasing post-COVID-19 due to the lack of income from tourism and volunteer programs. Nonetheless, our results reveal a form of resilience regarding the sanctuaries and the strategy applied to maintain their management. Consequently, we emphasize the specific impacts of the COVID-19 outbreak and its repercussions for conservation work. We discuss the difficulties that sanctuaries have faced throughout the crisis and present the best measures to prevent future outbreaks and protect biodiversity.

## 1. Introduction

The high proximity of wildlife to humans in landscapes they share leads to inevitable conflict [1,2,3]. Conflict with wildlife occurs due to the perception that risk is involved in interactions with wildlife, so steps are often taken to eliminate these animals [4,5,6]. Practitioners urge the development of coping mechanisms that allow humans and wildlife to thrive in a shared landscape [7,8,9]. Unfortunately, as habitats for wildlife diminish, overlap with humans becomes inevitable [10,11,12], consequently increasing the potential transmission of diseases and interspecies contact events [13].

The Pan Africa Sanctuary Alliance (PASA), a non-profit organization founded in 2000 in Entebbe, Uganda, is a coalition of a network of sanctuaries and wildlife centers across Africa, comprising 23 members. PASA defines its sanctuaries as places that “provide a safe and secure home for African apes and other primates in need. The welfare of the individual and the preservation of the species are considered equally important. Sanctuaries operate in the context of integrating approaches to conservation, which can include rehabilitation and reintroduction. The many different programs within the organization target the current issues causing the decline of primate species to secure a future for them, raising public awareness towards primate conservation through the pet trade, tourism, community involvement, and sustainable ecological projects. The sustainability of sanctuaries is often disturbed by external events such as disease outbreaks, and a model of resilience for these sanctuaries is necessary.

In December 2019, a virus called severe acute respiratory coronavirus 2 (SARS-CoV-2) started a pandemic worldwide, causing the coronavirus disease (COVID-19). Only a few cases, however, have been reported in Africa, which is surprising, as the continent contains 17% of the global population [14]. As the pandemic spread among humans, wildlife was indirectly and directly impacted, with cases linked to the human–animal interface, especially within tourism destinations [15]. The risk of transmission from interspecies events has been studied, and findings demonstrate that the protein sequence of the binding receptor to the virus SARS-CoV-2 is found in the genomes of most non-human primates; thus, they may also be infected by the virus [16]. The COVID-19 outbreak has made African sanctuaries and their inhabitants vulnerable. As there is no end in sight, the main objectives of PASA are to identify problems and build solutions through alliances, to preserve non-human primates, and ensure that they are given appropriate care. Meanwhile, the crisis has permitted researchers to analyze and demonstrate the effects of human activity restrictions on wildlife [17,18].

It has been shown in recent decades that viruses such as Ebola and Dengue have been transmitted from humans to hon-human primates [19,20]. Researchers still have not established whether the virus SARS-CoV-2 has the same impact on morbidity and mortality in great apes, our closest relatives, as it does in humans [21]. UNESCO reported for the first time, on 11 January 2020, that a gorilla (*Gorilla gorilla*) from the San Diego Zoo in California, USA, had tested positive for COVID-19. Whether in captivity or in the wild, great apes are genetically similar to us, and inevitably, a major pandemic such as COVID-19 may jeopardize their conservation. Multiple organizations and researchers have been evaluating measures to put into practice in order to reduce any risk of SARS-CoV-2 transmission to great apes [22]. In addition to primate research being paused or postponed, the IUCN Best Practice Guidelines for Health Monitoring and Disease Control in Great Ape Populations declared requirements that must be followed during observation or tourism involving great apes, and that must be thoroughly implemented on-site [23].

The illegal wildlife trade, which is still very active across Africa, continues to be a major concern for primates. Nijman et al. [24] stated that in urban markets, the live trade in primates, such as the pet-trade, involves hundreds of thousands of individuals a year, while the trade in primate parts amounts to millions a year, both internationally and nationally. In central Africa, the hunting of adult chimpanzees for meat can also result in orphaned chimpanzee young entering the pet trade, compounding ape decline [25]. Bushmeat consumption in many African countries results from a complex array of social, economic, and cultural reasons [26]. One of the factors that has increased the bushmeat trade is the urbanization of areas created by the logging sector, which has directly increased the demand for bushmeat consumption [27]. Each year, many primates, who are victims of illegal wildlife trafficking, enter the network of sanctuaries on the African continent. Many have a population flow of primates, where the influx is greater than the outflux [28]. African great apes, which include chimpanzees (*Pan troglodytes*), bonobos (*P. paniscus*), western gorillas (*Gorilla gorilla*), and eastern gorillas (*G. berengei*), are found in 21 countries across Africa. These species have a status of Endangered or Critically Endangered according to the IUCN Red List, and are threatened by trade despite the fact that it is illegal.

Here, we aim to identify the direct impacts of COVID-19 on PASA members and their efforts to maintain their primary goal of securing primate welfare. To avoid potential zoonosis transmission and ensure primates’ health, several sanctuaries had no other choice than to close their gates to visitors. In addition to the government-imposed lockdowns and decision-making, severe costs have been endured, and many sanctuaries are still trying to cope with these events. Our objective is to demonstrate the ways in which PASA have coped with these challenges post-COVID-19 to recover their sanctuaries and to establish recommendations to help sanctuaries working with vulnerable animal populations cope with future pandemics.

## 2. Materials and Methods

### 2.1. Data Collection

PASA was founded in 2000, with the aspiration of sharing and developing a network that would improve the captive welfare of primates confiscated from the trade, allowing sanctuaries to exchange resources and information. It has brought together 23 sanctuaries from 13 countries in Africa. PASA’s roles involve many conservation actions, such as providing the necessary funds for its members to thrive, establishing strategies to enhance primate care, increasing public awareness, and empowering the community. We used data from annual surveys voluntarily completed by PASA members since 2015. For the purpose of this paper, we only considered data from three time periods of COVID-19: before (2018–2019), at the start of (2019–2020), and during (2020–2021) the pandemic. These surveys are considered best practice and effective mechanisms of building a solid foundation and maintaining organization throughout all sanctuaries, based on their multidisciplinary roles that can ensure a future for primate species through their management within these facilities [29,30]. Each survey contained more than 30 questions concerning the sanctuary activities related to their management from July of the study year to June of the following year. From the overall questionnaire, we selected specific questions to achieve our aim (Table 1).

### 2.2. Data Analysis

To demonstrate the impact of COVID-19 on variables pertaining to PASA’s management, we divided the data into three time periods of COVID-19: before (2018–2019), at the start of (2019–2020), and during (2020–2021) the pandemic. We respected the disclosure agreement; hence, no sanctuary name is displayed to assure the privacy of PASA members. Additionally, the questions were voluntary, and some sanctuaries did not disclose information on their finances. From the questionnaire, we selected the following dependent variables: rescued primates, finances (revenue and expenses), and staff members employed. We ran Generalized Linear Mixed Models via the “glmmTMB” function in the “glmmTMB” package, as this function includes several fit families that are suitable for dealing with counts and zero-inflated distributions [31]. We used sanctuaries as random factors and included an ar1 covariance structure to consider the repeated measures (i.e., period) for each sanctuary. We tested the Poisson, genpoin, nbinom1, nbinom2 and Tweedie families and included or excluded a zero-inflation term based on the QQ plot residuals and residual vs. predicted plot from the package “DHARMa”. We ran pairwise contrasts using Bonferroni–Holm post hoc correction via the function “emmeans” in the package “emmeans”. For revenues, we ran a paired-samples t-test, as revenue data were only present in the 2020 and 2021 surveys, and the data were normally distributed. We considered *p* = 0.05 as significant. We ran all of the analyses using R v 4.1.0.

## 3. Results

### 3.1. Primate Arrivals

PASA sanctuaries care for 45 species/subspecies of primate, 17 of which are also regularly observed in the wild (Table 2). The number of individuals followed in the wild (previously released from sanctuaries or wild animals) decreased during the COVID-19 outbreak, especially red-eared guenons, mona monkeys, vervet monkeys, drills, and yellow baboons. For some species (i.e., mandrills and eastern chimpanzees), however, more individuals were regularly observed in the wild during than before the pandemic.

A total of 332 individuals were rescued before the pandemic, 238 individuals were rescued at the start, and 182 individuals were rescued during. The number of primates rescued by a sanctuary, however, did not vary in relation to COVID-19 (GLMM; χ^2^ = 0.37; *p* = 0.831). The estimated marginal means of primate arrivals for sanctuaries was 2.0 individuals (95% CI = 0.6–6.7 individuals) before the pandemic, 1.8 individuals (95% CI = 0.5–6.1 individuals) at the start, and 2.1 individuals (95% CI = 0.6–7.1 individuals) during. Of the total number of rescues, 48 (14.5%) before the pandemic, 49 (20.6%) at the start, and 81 (44.5%) during were carried out with the involvement of law enforcement.

### 3.2. Employees

The number of staff members employed by sanctuaries did not change significantly (Table 3). The majority of staff employed were African (before: 91.0% full time staff, 71.4% part-time staff; at the start: 90.8% full-time staff, 83.3% part-time staff; during: 90.6% full-time staff, 87.0% part-time staff).

### 3.3. Finances

Most sanctuaries (13 out of the 17 who completed the form) had a decrease in their revenues between the start of and during the pandemic. Overall, the revenues of the 17 sanctuaries significantly decreased from the start (USD 366,633.43 ± SE 60,997.15) to during (USD 290,020.61 ± SE 65,547.96) the pandemic (paired-samples *t*-test: t = 2.69, *p* = 0.016). Amongst the 23 PASA members, prior to the current pandemic, 18 offered volunteer programs, ten of which were main sources of income. Since the COVID-19 outbreak, ten sanctuaries had to completely shut down their volunteer programs, and eight only had a few visitors and volunteers. The sanctuaries changed their expenses depending on the COVID-19 period (GLMM; χ^2^ = 6.44; *p* = 0.040), but only had a tendency to have higher expenses at the start (EMM = USD 302,145; 95% CI = USD 216,366–421,932) of the COVID-19 outbreak compared to before COVID-19 (EMM = SDU 256,513; 95% CI = USD 182,369–360,802) (Bonferroni–Holm post hoc; *p*-value = 0.063). Expenses in the period during the COVID-19 outbreak were not different from those during the other periods (EMM = USD 276,964; 95% CI = USD 198,334–386,768).

### 3.4. Funding Opportunities

The percentages of sanctuaries that indicated the need for additional funds for the improvement of enclosures, the protection of wild populations, education, and community development were reduced during COVID-19 compared to before and at the start of COVID-19 (Figure 1). More sanctuaries indicated the need for funds to build new enclosures or expand current enclosures at the start of and during COVID-19. More sanctuaries also indicated the need to have money to save in a reserve fund at the start and during COVID-19. In 2021, we also asked members if they were satisfied with the current funding available, and six responded with “satisfied”, four with “not at all satisfied”, and the rest with “partially satisfied”.

## 4. Discussion

Our results show that our variables related to sanctuary management, such as primate arrivals, employees, and finances, were not significantly affected by COVID-19. Nonetheless, the revenues for several captive facilities showed a significant decrease pre- and post-COVID-19, which cannot be ignored. Overall, their management has been maintained with a sense of duty to preserve the health of captive primate populations; funding opportunities and sources of recovery funds might be the reasons for such stability. Wildlife conservation organizations provide benefits to many African countries, but COVID-19 has brought additional challenges to an unstable Africa regarding shared conservation strategies [2,32]. Ecosystems and biodiversity have been shaken by the COVID-19 outbreak [33]. The pandemic has established many restrictions regarding conservation research [34], with fieldwork areas close to non-human primates being restricted due to sanitary and health procedures. Some have suggested that the ideal recommendation is to postpone every research project until further notice to improve the security of primate populations. Although it has been suggested that the COVID-19 outbreak and its following lockdowns gave back nature’s rights and enhanced environmental quality [35,36,37], worldwide perspectives on COVID-19’s impact on wildlife conservation suggest otherwise [38,39]. Although many celebrated the lack of humans in wildlands during the pandemic, poaching and hunting from wildlife trafficking networks also increased, giving no choice to local populations but to return to their old livelihoods [40]. 

According to our results, primate populations entering sanctuaries did not decrease during COVID-19, and the proportion of rescue efforts, in collaboration with the involvement of law enforcement, increased, which is essential to the protection of wild populations [41]. In many cases, parks and sanctuaries suffered from a lack of income generated by the tourism industry and travel restrictions; moreover, health measures required many to shut down their volunteer programs, and reductions in grants limited staff on-site [42]. As finances ran low, with income within facilities as identified in our research, ecotourism appears to have been an important source of funding in this area [43]. By definition, ecotourism has more than one function; according to The International Ecotourism Society, its purpose is to increase opportunities for responsible travel to a preserved environment, to link communities with visitors, and to enhance education through experimental conservation explanation and interpretation, thus creating awareness and protecting our futures [44]. National and international tourism were ceased as a response from the government to contain the pandemic; therefore, sanctuaries have found themselves deprived of their natural sources of income [45]. In addition, the threat of potential COVID-19 transmission to primates in sanctuaries interferes greatly with their management; as a matter of fact, almost all volunteer programs were shut down during the pandemic, and these represent an important source of income for many PASA members.

The essential management of sanctuaries includes many aspects of conservation, including rescue, rehabilitation, and release, which were not affected directly by the COVID-19 outbreak. Threats to primates, such as the illegal wildlife trade, however, increased after COVID-19, leaving local communities to enhance their livelihoods through hunting; this is also linked to fewer job opportunities that researchers, for example, can provide during fieldwork [17]. In addition, natural habitats emptied of tourism created more hunting opportunities and participation in wildlife trafficking. The effect followed by the imposed lockdown caused a lack of external people within sanctuaries, such as visitors, volunteers, and researchers.

Benefits also arose from the COVID-19 outbreak, opening new areas for conservation research to acknowledge its impact, particularly in tourist sectors. Some new research has been conducted on protected African wildlife areas and the impact of the pandemic on conservation [46,47]. The direct effect of COVID-19 on wildlife conservation in Africa is linked to its shrinking economy, with funding for conservation declining and directly impacting the effectiveness of management within conservation areas [48]. Cumming et al. [49] established future recommendations for coping with the pandemic financially in conservation areas. To summarize, adaptation to the COVID-19 outbreak demonstrates an important form of resilience from captive facilities and the wide range of decisions taken to improve conservation efforts in the post-COVID-19 era [50,51]. To cope with the pandemic, many had to find alternative ways to enhance ecotourism and rise through a difficult time [52], while also fighting to avoid the collapse of the local economy [53].

Wildlife sanctuaries play a vital role in conservation in primate range countries, and understanding the impact of COVID-19 on their ability to cope during a global crisis is vital. Our study highlights the importance of African sanctuaries in conservation. PASA members allowed us to assess their perspectives towards the pandemic’s impact, in the hope of informing future research into sanctuary management and identifying strengths and weaknesses through a crisis period. The potential outcome of this study could be the design of a worldwide sanctuary management resilience program in light of the pandemic, building the entrance in a new kind of conservation approach post-COVID-19 to protect biodiversity and raise awareness among the wider public.

## 5. Conclusions

We demonstrated that COVID-19 impacted African sanctuaries, especially regarding the reduction in funding, leading to difficulties in ensuring care for non-human primates within facilities. If many sanctuaries (the ones that do not/marginally rely on ecotourism) managed to develop a resilient post-COVID-19 approach, the way to complete recovery is long, as the post-COVID-19 period could be longer than anticipated. Consequently, they need help, including crucial identification of all the weaknesses in their management during COVID-19 and a broader source of income to ensure the continued care of sanctuary inhabitants. From our study and the overall answers to the annual survey, we provide suggestions on best practices for sanctuaries and guidance for PASA members in the post-COVID-19 period (Table 4). These best practices can be used by other organizations of sanctuaries.

## Figures and Tables

**Figure 1 animals-13-01486-f001:**
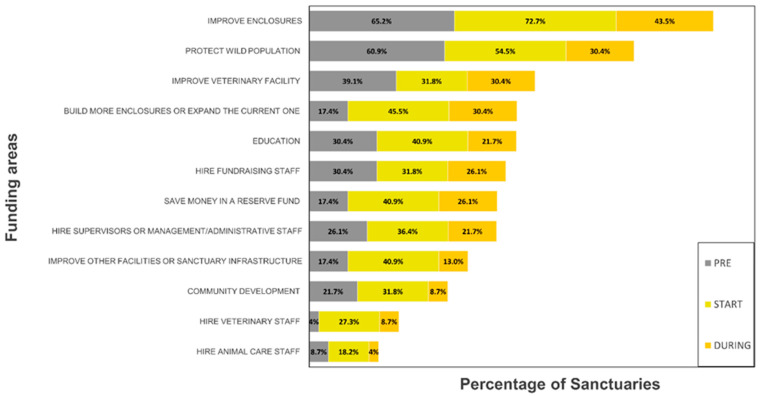
Areas voted for by sanctuaries as percentages in response to the question “If additional funding became available to you, how would you use it?”, according to the three COVID-19 periods: before (2018–2019), at the start of (2019–2020), and during (2020–2021).

**Table 1 animals-13-01486-t001:** Selected questions relevant to this study from Pan African Sanctuary Alliance annual survey.

Survey Questions from PASA Members
How many animals do you care for at your sanctuary now? How many are observed in the wild? At your sanctuary, how many new individuals did you acquire? How many were rescued with and without the involvement of law enforcement? How many part-time and full-time staff members (international and local) do you employ? In your last financial year, what was your total revenue? Total of expenses? If additional funding became available to you, how would you use it? (max. 3 answers) Do you offer volunteer opportunities at your centre? How has the coronavirus pandemic impacted your volunteer and visitor program? Are volunteers/visitors an important source of revenue for your organization?

**Table 2 animals-13-01486-t002:** Total number of individuals for each primate species present in PASA sanctuaries or regularly observed in the wild by PASA members before COVID-19 outbreak (June 2019). Number in parenthesis indicates the variation during COVID-19 outbreak (June 2021).

Species	Common Name	Sanctuary	Wild
*Allenopithecus nigroviridis*	Allen’s swamp monkey	1	0
*Allochrocebus lhoesti*	L’Hoest’s monkey	10 (+2)	0
*Allochrocebus preussi*	Preuss’s guenon	1	0
*Cercocebus agilis*	Agile mangabey	33 (−4)	0
*Cercocebus torquatus*	Red-capped/white-collared mangabey	13	0
*Cercocebus/Lophocebus* spp.	Mangabey (unknown species)	0 (+11)	0
*Cercopithecus ascanius*	Red-tailed monkey/red-tailed guenon	14 (+5)	0
*Cercopithecus cephus*	Mustached guenon	11 (+8)	0 (+2)
*Cercopithecus denti*	Dent’s mona monkey	2	0
*Cercopithecus erythrotis camerunensis*	Red-eared guenon	8 (−5)	30 (−28)
*Cercopithecus hamlyni*	Hamlyn’s monkey/owl-faced guenon	17 (−1)	0
*Cercopithecus mitis albogularis*	Sykes’ monkey/white-throated guenon	10 (−7)	0 (+5)
*Cercopithecus mitis labiatus*	Samango monkey/blue monkey	0 (+4)	0
*Cercopithecus mitis* ssp.	Blue monkey (unknown subsp.)	19 (−1)	3 (−3)
*Cercopithecus mona*	Mona monkey	20 (+2)	65 (−62)
*Cercopithecus neglectus*	De Brazza’s monkey	7 (−2)	0
*Cercopithecus nictitans*	Putty-nosed/greater spot-nosed monkey	27 (+4)	0
*Cercopithecus pogonias pogonias*	Crowned guenon/crested mona monkey	2	0
*Cercopithecus sclateri*	Sclater’s guenon	10	0
*Cercopithecus* spp.	Guenon (unknown sp.)	6 (+2)	0
*Chlorocebus cynosuros*	Malbrouck monkey	13 (+5)	0
*Chlorocebus pygerythrus*	Vervet monkey	707 (+23)	200 (−159)
*Chlorocebus tantalus*	Tantalus monkey	48 (−38)	0
*Colobus angolensis palliatus*	Angolan black-and-white colobus	0 (+1)	1 (−1)
*Erythrocebus patas*	Patas monkey	10	0
*Galago* sp.	Lesser bushbaby/galago (unknown sp.)	0 (+1)	0
*Gorilla gorilla gorilla*	Western lowland gorilla	66 (−1)	63 (−1)
*Lophocebus albigena*	Grey-cheeked mangabey	7 (−4)	0
*Lophocebus aterrimus*	Black-crested mangabey	1	0
*Mandrillus leucophaeus*	Drill	643 (+36)	58 (−55)
*Mandrillus sphinx*	Mandrill	66 (+4)	289 (+222)
*Miopithecus ogouensis*	Gabon talapoin/northern talapoin monkey	3	0
*Otolemur crassicaudatus crassicaudatus*	Greater bushbaby/galago	4 (−1)	5
*Otolemur sp.*	Greater bushbaby/galago (unknown sp.)	1 (−1)	0
*Pan paniscus*	Bonobo	74 (+6)	13 (+3)
*Pan troglodytes ellioti*	Nigeria–Cameroon chimpanzee	72 (−5)	3 (−2)
*Pan troglodytes schweinfurthii*	Eastern chimpanzee	251 (+17)	150 (+100)
*Pan troglodytes troglodytes*	Central chimpanzee	376 (−118)	11 (+1)
*Pan troglodytes verus*	Western chimpanzee	264 (+13)	7 (−7)
*Pan troglodytes* ssp. (hybrid)	Hybrid chimpanzee	30 (+1)	0
*Pan troglodytes* ssp. (unknown)	Chimpanzee (unknown sp.)	110 (+149)	0
*Papio anubis*	Olive baboon	74 (−14)	0
*Papio cynocephalus*	Yellow baboon	38 (+14)	60 (−60)
*Papio kindae*	Kinda baboon	0 (+1)	0
*Perodicticus potto*	Potto	0 (+1)	0
**Total**		**3081 (+96)**	**958 (−45)**

**Table 3 animals-13-01486-t003:** Number of full-time (FT) and part-time (PT) local and international staff members of PASA sanctuaries (N = 23) in relation to COVID-19 periods: before (2018–2019), at the start (2019–2020), and during (2020–2021). Numbers are estimated marginal means (EMM) and 95% confidence intervals (CI) based on Generalized Linear Mixed Models.

	Before			At the Start			During			GLMM
	EMM	95% CI	Total	EMM	95% CI	Total	EMM	95%CI	Total	
African FT	25.6	17.9–36.6	756	24.5	17.1–35.0	623	25.2	17.6–36.0	738	χ^2^ = 1.1, *p* = 0.590
African PT	0.5	0.1–1.9	35	0.9	0.3–2.6	45	0.7	0.2–2.2	40	χ^2^ = 1.3, *p* = 0.535
International FT	1.7	0.8–3.5	75	1.7	0.8–3.5	63	1.8	0.9–3.5	77	χ^2^ = 0.0, *p* = 0.993
International PT	0.2	0.0–1.5	14	0.1	0.0–0.9	9	0.1	0.0–0.8	6	χ^2^ = 0.9, *p* = 0.628

**Table 4 animals-13-01486-t004:** Best practices for sanctuaries and guidance for PASA in the post-COVID-19 period.

Best Practice	Description
Members	The creation of bonds between PASA members; it is vital to build a united front.
Information sharing	Workshops and meetings should be planned in order to share strategies and improvements within sanctuaries.
Funding	Reach for broader sources of funding. The COVID-19 outbreak revealed that relying on one main source of income represents a risk [54].
Social Media	Recruit remote volunteers for social media coordination. Social media is a platform that can be used to reach a wider public, and therefore, to spread awareness and obtain worldwide donations.
Convivial conservation	Implement the “built on the politics of equity, structural change, and environmental justice” notion from Garber [55].
Conservation education	Empower local communities with conservation actions. Educational conservation should thrive and promote local volunteering.
Priorities	Prioritize investments by identifying the most concerning areas requiring financial support.
Emergency plan	Establish an emergency plan concerning the maintenance of a future crisis, to act effectively and safely, hence assuring primate welfare.
Contingency plan	Keep human and non-human primates safe, and assure locals and internationals of their own safety within the facilities.
Management training	For efficiency, it is important to train staff within sanctuaries to cover different roles in times of crisis.

## Data Availability

The data presented in this study are available upon request from the corresponding author.

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
