# Peer review of "COVID-19’s Impact on the Pan African Sanctuary Alliance: Challenging Times and Resilience from Its Members"

_animals, 2023, doi:10.3390/ani13091486_

Round 1

Reviewer 1 Report (Previous Reviewer 1)

No other comments. 

Author Response

Thanks

Reviewer 2 Report (New Reviewer)

This is a timely and useful piece of work that brings together evidence to support anecdotal reports of the effect of the pandemic on animal charities.

The work is is sound and well written and I only have a few minor points and comments.

The method compares survey responses completed since 2015  but includes a question relating on impacts of the coronavirus. When was this asked? It should be made clear what years this question refers to. 
The surveys were completed voluntarily and not all questions answered, how many sanctuaries complete surveys each year (this should be included at the start of the results). N should be stated per question.

line 646-known a rise-grammar

line 707 what is a thick box? 

table 4 and figure 2 could be combined for ease or consider alternative icons as prioritise, emergency and contingency are less easily recognisable. 
references 54 and 55 do not appear in text.

Author Response

This is a timely and useful piece of work that brings together evidence to support anecdotal reports of the effect of the pandemic on animal charities.

The work is is sound and well written and I only have a few minor points and comments.

We thank the reviewer for the kind words and final suggestions

The method compares survey responses completed since 2015  but includes a question relating on impacts of the coronavirus. When was this asked? It should be made clear what years this question refers to. 

We clarified the data collection period

The surveys were completed voluntarily and not all questions answered, how many sanctuaries complete surveys each year (this should be included at the start of the results). N should be stated per question.

All the sanctuaries voluntarily answered most of the questions. When the question was not answered by some we included the information (see section 3.3)

line 646-known a rise-grammar

Edited

line 707 what is a thick box? 
removed

table 4 and figure 2 could be combined for ease or consider alternative icons as prioritise, emergency and contingency are less easily recognisable. 

We agree, we removed fig 2 as redundant and not so informative
references 54 and 55 do not appear in text.

They are in table 4

This manuscript is a resubmission of an earlier submission. The following is a list of the peer review reports and author responses from that submission.

Round 1

Reviewer 1 Report

Authors provides interesting information on management and conservation challenges coping by sanctuaries. Overall, the manuscript is well written and data presentation clear. However, simple syntax errors such as too long sentences are present throughout the text making it sometimes difficult to read. In particular the entire Introduction paragraph need to be rephrased in order to make concepts presented linear and easier to read (a possible example is provided in the .pdf attached file).

In addition, I suggest avoiding the use of personal language and replaced with the use of impersonal phrase, passive verb or change of subject. Some examples:

L13. “We analyzed the responses of the 23 sanctuaries members of the Pan African Sanctuary Alliance (PASA) to the Covid-19 pandemic, analyzing the periods pre, start, and during the pandemic in terms of primates rescued, employees, and expenses.”

can be rephrased as

“The responses of the 23 sanctuaries members of the Pan African Sanctuary Alliance (PASA) to the Covid-19 pandemic were analysed. Responses were analysed in the periods pre, start, and during the pandemic in terms of primates rescued, employees, and expenses.”

Concepts provided in the introduction section need to be better organized to make it easy to read (a possible example of text reorganization is provided in the pdf. attached file)

L. 27-29 “We collected data from PASA annual surveys for its members and ran Generalized Linear Mixed Models to highlight any significant differences in their management that could have been linked to Covid-19”

 can be rephrased as

“Data were collected from PASA annual surveys for its members. Collected data were analysed by Generalized Linear Mixed Models to highlight any significant differences in PASA management potentially linked to Covid-19” 

L148-149: “From the questionnaire, we selected the following dependent variables:”

can be rephrased as

“From each questionnaire the following dependent variables were selected:”

 Some other minor comments are provided in the .pdf attached file

Reviewer 2 Report

The concept of the authors is good, but the methods and content are not convincing/sufficient to publish it at this stage. 

Reviewer 3 Report

Reviewer Comments

The manuscript is titled: “Covid-19 impact on the Pan African Sanctuary Alliance: challenging times and resilience from its members”. It is a Research article which focuses on effects of covid-19 on conservation. The manuscript is publishable after major revision ‘especially the introduction). There are some imperfections which must be improved before the acceptance of this manuscript, such as:

I. The introduction is not well presented since there is some sections missed

II. The manuscript lacks clear hypotheses, objectives and research questions. At least one of these three elements should be there.

III. The discussion talked about some results which are not found in the manuscript (line Some Issues262-268)

1. Line 67: malaria is not a disease caused by a virus, please change this.

2. Lines 84-93: this paragraph is not essential in the introduction

3. Line 127: author could use only PASA since it has been written in full before.

4. Line 141: that table must be in index

5. Line 262-268: I don’t understand how authors did know that, as such kind of affirmations are not in their results because there is no question about that.